# Familial Partial Lipodystrophy—Literature Review and Report of a Novel Variant in *PPARG* Expanding the Spectrum of Disease-Causing Alterations in FPLD3

**DOI:** 10.3390/diagnostics12051122

**Published:** 2022-04-30

**Authors:** Lena Rutkowska, Dominik Salachna, Krzysztof Lewandowski, Andrzej Lewiński, Agnieszka Gach

**Affiliations:** 1Department of Genetics, Polish Mother’s Memorial Hospital-Research Institute, 93-338 Lodz, Poland; dominik.salachna@iczmp.edu.pl; 2Department of Endocrinology and Metabolic Diseases, Medical University of Lodz, 90-419 Lodz, Poland; 3Department of Endocrinology and Metabolic Diseases, Polish Mother’s Memorial Hospital-Research Institute, 93-338 Lodz, Poland

**Keywords:** inherited lipodystrophy, familial partial lipodystrophy type 3, metabolic disorder, lipids, *PPARG* gene, genetic background, genotype-phenotype correlation

## Abstract

Familial partial lipodystrophy (FPLD) is a rare genetic disorder characterized by the selective loss of adipose tissue. Its estimated prevalence is as low as 1 in 1 million. The deficiency of metabolically active adipose tissue is closely linked with a wide range of metabolic complications, such as insulin resistance, lipoatrophic diabetes, dyslipidemia with severe hypertriglyceridemia, hypertension or hepatic steatosis. Moreover, female patients often develop hyperandrogenism, hirsutism, polycystic ovaries and infertility. The two most common types are FPLD type 2 and 3. Variants within *LMNA* and *PPARG* genes account for more than 50% of all reported FPLD cases. Because of its high heterogeneity and rarity, lipodystrophy can be easily unrecognized or misdiagnosed. To determine the genetic background of FPLD in a symptomatic woman and her close family, an NGS custom panel was used to sequence *LMNA* and *PPARG* genes. The affected patient presented fat deposits in the face, neck and trunk, with fat loss combined with muscular hypertrophy in the lower extremities and hirsutism, all features first manifesting at puberty. Her clinical presentation included metabolic disturbances, including hypercholesterolemia with severe hypertriglyceridemia, diabetes mellitus and hepatic steatosis. This together with her typical fat distribution and physical features raised a suspicion of FPLD. NGS analysis revealed the presence of missense heterozygous variant c.443G>A in exon 4 of *PPARG* gene, causing glycine to glutamic acid substitution at amino acid position 148, p.(Gly148Glu). The variant was also found in the patient’s mother and son. The variant was not previously reported in any public database. Based on computational analysis, crucial variant localization within DNA-binding domain of PPARγ, available literature data and the variant cosegregation in the patient’s family, novel c.443G>A variant was suspected to be causative. Functional testing is needed to confirm the pathogenicity of the novel variant. Inherited lipodystrophy syndromes represent a heterogenous group of metabolic disorders, whose background often remains unclear. A better understating of the genetic basis would allow earlier diagnosis and targeted treatment implementation.

## 1. Introduction

Lipodystrophy syndromes are a heterogeneous group of genetically inherited or acquired conditions, characterized by dysfunctional white adipose tissue. They are considered ultra-rare syndromes with an estimated prevalence of 1.3–4.7 cases per million [1]. Based on the degree of fat loss we distinguish partial and generalized lipodystrophy [2]. There are four major categories of lipodystrophy syndromes: Congenital Generalized Lipodystrophy (CGL), Familial Partial Lipodystrophy (FPLD), Acquired Generalized Lipodystrophy (AGL) and Acquired Partial Lipodystrophy (APL). The two most prevalent subtypes of genetic lipodystrophies are Congenital Generalized Lipodystrophies and Familial Partial Lipodystrophies. The main differentiating criteria are molecular etiology and pattern of adipose tissue distribution [3]. Generally, there are four types of CGL, seven types of FPLD and a few other unclassified forms [1]. The described prevalence of CGL is approximately 1 in 10 million and for FPLD about 1 in 1 million but it may be underestimated [3].

This paper reports a novel c.443G>A variant probably affecting the DNA-binding domain of the PPARγ receptor in a symptomatic 29-year-old index patient and family. The presented literature review is focused mainly on FPLD type 2 and 3, as this was the scope of clinical consideration in our case.

## 2. Familial Partial Lipodystrophy (FPLD)

Familial Partial Lipodystrophy is a rare genetic disorder usually characterized by selective loss of adipose tissue in the extremities and gluteal region, without any change in abdominal and visceral fat. In most cases, abnormal fat distribution becomes apparent at puberty [4]. This selective deficiency of metabolically active adipose tissue is tightly linked with a wide range of metabolic complications, such as insulin resistance, lipoatrophic diabetes, dyslipidemia with severe hypertriglyceridemia, hypertension or hepatic steatosis. Moreover, female patients often develop hyperandrogenism, hirsutism, polycystic ovaries and infertility [5]. The extent of fat loss often determines the severity of metabolic consequences. For example, patients with generalized lipodystrophies have more severe diabetes, hypertriglyceridemia, or hepatic steatosis than those with partial lipodystrophies.

There are seven subtypes of FPLD (characterised in Table 1)-six caused by mutations in various genes (*LMNA*, *PPARG*, *PLIN1*, *CIDEC*, *LIPE*, *AKT2* or *CAV1*); the condition can be inherited in either a dominant (mostly) or recessive manner.

Variants within the *LMNA* and *PPARG* genes account for more than 50% of all reported FPLD cases [3]. Both genes play a crucial role in the differentiation and proper functioning of adipose tissue [6]. The most common form of FPLD is type 2 (Dunnigan type; OMIM #151660), which is inherited in an autosomal dominant manner [1].

### 2.1. Familial Partial Lipodystrophy Type 2 (Dunnigan Type)

The Dunnigan-type familial partial lipodystrophy is caused by mutations in the *LMNA* gene, located on the long arm of chromosome 1 (1q21–q22). As the *LMNA* gene is ubiquitously expressed, different mutations throughout the gene can lead to at least 14 diseases from various forms of muscular dystrophy to dilated cardiomyopathy [6,7]. Disorders associated with *LMNA* aberrations are collectively described as laminopathies.

The *LMNA* gene encodes A-type nuclear lamins produced via alternative splicing. The two major isoforms, sharing the first 566 amino acids, are lamin A and C (Lamin A/C) [8]. They are primarily localized below the inner nuclear membrane and form part of the nuclear lamina [8]. Lamin A/C interacts with the cytoskeleton and provides structural stability for the nuclear envelope [6].

At present, it is unclear how unique *LMNA* mutations can cause an adipose tissue-specific disease like FPLD2 [8], as well as its late manifestation [7]. It is suspected that the underlying cause of disease is altered cell division, increased apoptosis and cell death, due to disrupted lamin-chromatin interactions [6]. *LMNA* mutations are thought to induce structural modifications of nuclear lamina, resulting in cytotoxic accumulation of immature proteins and therefore probably weakness of nuclear lamina bonds [9]. Approximately 90% of *LMNA* mutations seen in FPLD2 are localized to exon 8 [8], which encodes the C-terminal domain of lamin A/C. The most frequent mutation is Arg482Gln [4], resulting in arginine to glutamine replacement within a highly-conserved region across the species [10]. A recent study from 2020 reported that variants Arg482Trp/Gln are responsible for 80% of FPLD2 cases [11].

FPLD2 is characterized by the loss of subcutaneous fat in the extremities and trunk and its accumulation on the neck, submental regions, supraclavicular area and face (“Cushingoid appearance”) [12,13]. The distribution of adipose tissue appears normal at birth and during childhood and becomes apparent at the onset of puberty [12]. The loss of almost all subcutaneous adipose tissue results in a characteristic phenotype of “increased muscularity” in the arms and legs. The specific pattern of phenotypic features is more recognizable in women than in men [12]. Patients with FPLD2 develop a multitude of metabolic complications such as insulin resistance, hypertriglyceridemia, hepatic steatosis and others, as described in Section 3. A less aggressive metabolic profile is reported in affected males [13]. FPLD2 patients commonly develop cardiovascular diseases and myopathies to varying degrees [13]. Cardiomyopathies induced by *LMNA* gene mutations are characterized by a sudden and aggressive clinical course. They can lead to unexpected cardiac death at earlier ages than in other familial cardiomyopathies [14].

### 2.2. Familial Partial Lipodystrophy Type 3

The molecular basis of FPLD3 is loss-of-function mutations in the *PPARG* gene, which is located on the short arm of chromosome 3 (3p25.2). The *PPARG* gene encodes a member of the peroxisome proliferator-activated receptor (PPAR) subfamily of nuclear receptors. PPAR nuclear receptors have three isoforms (PPARα, PPARδ, and PPARγ) with different tissue distribution and biological functions. Peroxisome proliferator-activated receptor gamma (PPARγ) is a key regulator of adipocyte differentiation, distribution and function [13], mediating in glyceroneogenesis, lipolysis, lipid uptake, synthesis and storage [15]. It is highly expressed in white (WAT) and brown adipose tissue (BAT) [16]. It is suspected that mutated *PPAR*γ inhibits the adipocyte differentiation taking place during adipogenesis. As a result, the fatty tissue loses its ability to correctly synthesise and store triglycerides to free fatty acids and glycerol from stored triglycerides in postresorptive and starvation states, and biosynthesise and secret adipokines.

PPARs form heterodimers with retinoid X receptors (RXRs), which regulate transcription of various PPAR-responsive genes. There are no other promoters that can activate autonomously adipogenesis in the absence of *PPARG* [9].

The *PPARG* gene contains nine exons (A1, A2, B, 1, 2, 3, 4, 5 and 6), that may create four PPARγ mRNA isoforms, as a result of various promoter sites and alternative splicing. Transcripts PPARγ1, γ3 and γ4 lead to PPARγ1 protein synthesis, while transcript PPARγ2 encodes the PPARγ2 protein. PPARγ1 protein is found in most human tissues, while PPARγ2 predominantly occurs in adipose tissue [17].

The PPARγ protein is composed of four functional domains, of which the most essential are DBD (DNA-binding domain) and LBD (ligand-binding domain). The description of each PPARγ domain is presented in Table 2. The centrally-located DBD domain is highly conserved among species and between nuclear receptors [15], hence DBD mutants demonstrate less efficient DNA binding and can significantly reduce *PPARG* transcriptional activity. Next, LBD is the largest and second most conserved domain among nuclear receptors, after the DNA-binding domain [18]. It enables the binding of large hydrophobic particles, such as polyunsaturated fatty acids (arachidonic acid, linoleic acid, linolenic acid) and their metabolic products [17]. Aberrations within LBD binding pocket can lead to incorrect ligand attachment and therefore inhibit the activation of PPARγ receptor.

The clinical features of type 3 lipodystrophy are similar or sometimes less prominent than those of FPLD2. As noted by Vasandani et al. higher total fat occurs in FPLD2 than FPLD3 patients (26.1% vs. 21.6%) with higher triceps skinfold thickness (11.3 mm vs. 5.8 mm) [19]. FPLD3 patients are more likely to demonstrate loss of subcutaneous fat in the lower limbs and distal upper limbs [13]. Moreover, FPLD3 is characterised by early-onset hypertension, that can discriminate FPLD3 from FPLD2 [7].

## 3. Metabolic Abnormalities in Lipodystrophy

### 3.1. Insulin Resistance (IR) and Diabetes Mellitus

The core metabolic feature characterizing basically all lipodystrophy syndromes, is insulin resistance [2]. The presence of insulin resistance induces the development of diabetes, hypertriglyceridemia, polycystic ovary syndrome (PCOS) and non-alcoholic fatty liver disease [20].

The inability to maintain proper fat storage in adipose tissue leads to failure of buffering postprandial lipids. Secreted adipokines induce excessive levels of triglycerides and lipid intermediates in the circulation [21]. Excess triglycerides cannot be stored in adipose tissue, which results in their deposition in ectopic sites, such as liver or skeletal muscles [12]. The lipotoxicity of this mechanism probably induces development of insulin resistance. The severity of IR is broadly proportional to the extent of alteration within adipose tissue [2]. Diabetes mellitus and/or insulin resistance was identified in 51.8% of partial lipodystrophy patients in one study [13]. It has also been found that diabetes mellitus is more likely in FPLD3 than FPLD2 (72% vs. 44%) [19]. Interestingly, women are more likely to be affected than men (above 50% vs. 20%) [6].

One of the cardinal features marking severe IR is **acanthosis nigricans** (AN). The condition is characterized by hyperkeratosis, sometimes with hyperpigmentation, typically most prominent in body flexures [22]. A high insulin level in the bloodstream stimulates keratinocytes and fibroblasts to more potent growth and proliferation, which underlies the process of AN formation [23].

### 3.2. Hyperlipidemia with Hypertriglyceridemia

Dyslipidemia, which is found in most types of lipodystrophy is characterized by marked hypertriglyceridemia and reduced HDL cholesterol levels. The severity of lipid abnormalities reflects the degree of body fat reduction [24] and is strictly associated with prevalent forms of insulin resistance [22].

Marked hypertriglyceridemia is thought to be the first lipid indicator of ongoing lipodystrophy [25]. It is clear that absence of normal fat distribution disrupts correct lipid homeostasis. The underlying cause of a very high triglyceride level is probably increased VLDL synthesis from the fatty liver and reduced clearance of TG-rich lipoproteins [24]. Therefore, the presence of severe hypertriglyceridemia (>500 mg/dL) nonresponsive to medical therapy, should raise a suspicion of lipodystrophy. Extreme hypertriglyceridemia occurs also in uncontrolled diabetes; however, restoring glycemic control results in regaining body fat [19], which can be part of the differential diagnosis.

The exact pathogenesis of insulin resistance and hyperlipidemia occurring in congenital lipodystrophies is largely unknown [6]. It is estimated that 77.2% of FPLD and 70% of CGL patients display severe hypertriglyceridemia [13,24]. FPLD3 patients are more likely to demonstrate high triglyceride levels than those with FPLD2 (84% vs. 66%) [19]; the condition may be accompanied by a history of pancreatitis directly correlated with moderate to extreme TG levels [26]. The more frequent hypertriglyceridemia observed in FPLD3 patients is accompanied by a higher risk of acute pancreatitis compared to FPLD2 patients (52% vs. 13%) [19]. Interestingly, triglyceride levels are about 2–3 times higher in females than in males among FPLD patients [24]. It has been demonstrated that administration of recombinant human methionyl leptin (meterleptin) results in 60% decrease in triglycerides, with no influence on HDL concentration [25].

The presence of metabolic dyslipidemia (high triglycerides and low HDL cholesterol), as a consequence of ectopic adipose tissue storage, can lead to **non-alcoholic fatty liver disease** (**NAFLD**). NAFLD encompasses non-alcoholic simple steatosis (SS), which may progress to non-alcoholic steatohepatitis (NASH), then fibrosis and NASH-related cirrhosis [27]. The main components of dyslipidemia and its possible health consequences are presented in Figure 1. The severity of NAFLD may depend on the type of lipodystrophy, but also on the specific mutation in the relevant gene. For example, patients with *PPARG* mutation (FPLD3) present more severe hepatic steatosis than those with *LMNA* mutations (FPLD2) [27].

## 4. Results

### 4.1. Clinical Characteristics of Index Patient

The index patient was a 29-year-old woman with characteristic signs of partial lipodystrophy, including fat deposits in the face, neck and trunk, fat loss combined with muscular hypertrophy in the lower extremities and hirsutism. All described features became visible at puberty. At the age of 18, the patient was admitted to hospital with a triglyceride level of 1700 mg/dL and total cholesterol 400 mg/dL. The patient has also provided test results documenting high lipid parameters in her early teenage years. Implemented treatment with Lipanthyl (generic name-fenofibrate) and Roswera (generic name-rosuvastatin) did not bring the expected decrease of lipid levels. Therefore, the patient stopped taking her medication at the age of 20. The oral glucose tolerance test (OGTT; 0′—105 mg/dL, 60′—232 mg/dL, 120′—217 mg/dL) performed during pregnancy revealed diabetes, so high-dose insulin therapy was administered.

At the age of 25, the patient started treatment with 75 µg of L-thyroxine. Further laboratory tests showed persistent mixed dyslipidemia with total cholesterol 299 mg/dL, LDL cholesterol 111 mg/dL, HDL cholesterol 25 mg/dL and TG 870 mg/dL. After discontinuation of the metformin treatment, the OGTT test (0′—93 mg/dL, 60′—222 mg/dL, 120′—181 mg/dL) demonstrated impaired glucose tolerance with biochemical features of high cellular insulin resistance. The Insulin Resistance Index (IRI) was 1.85. The HOMA-IR parameter was 6.43. The Hemoglobin A1c was 36 mmol/mol and 5.43%. A therapy of metformin, pioglitazone and fenofibrate was implemented. Abdominal ultrasonography imaged features of hepatic steatosis.

The family history revealed the same pattern of subcutaneous fat loss and muscular hypertrophy with diabetes and hypertriglyceridemia in the patient’s mother. Marked hypertriglyceridemia was identified in the patient’s son at the age of three years. His physical appearance is not marked by lipodystrophy so far. The boy is also not receiving any pharmacological treatment. The appropriate treatment will be implemented in accordance with latest recommendation described in detail in the attached citation [28]. The mother’s sister died of a heart attack aged 40. She had presented a characteristic lipodystrophy phenotype including fat deposits in the face, neck and trunk and fat loss combined with muscular hypertrophy in the lower extremities. Grandfather died at the age of 71 with a diagnosis of cancer. No data was available on dyslipidemia or other metabolic abnormalities. Great-grandmother had a distinctive physical appearance suggestive of partial lipodystrophy with muscular legs and arms and an accumulation of subcutaneous fat on her face, trunk and abdomen. She died at the age of ninety.

The index patient, her affected mother and 3-year-old son were referred for genetic testing. Their lipid profiles are presented in Table 3.

### 4.2. PPARG Mutation

We identified a novel missense heterozygous variant c.443G>A in exon 4 of *PPARG* gene, causing glycine to glutamic acid substitution at amino acid position 148, p.(Gly148Glu). The variant was found in the index patient, her affected mother and son. The variant was not previously reported in the HGMD (Human Gene Mutation Database), ClinVar and LOVD (Leiden Open Variation Database 3.0) with no record in known population genetic databases such as ExAC, gnomAD or 1000 Genomes Project. Based on ACMG–AMP criteria it was assigned to class 4, likely pathogenic.

The mutation is located in the highly conserved DBD domain (Figure 2a) within the first zinc finger motif, a structure involved in DNA binding (Figure 2d). The homology modelling visualised an amino acid substitution (G → E) resulting in the presence of an additional side chain of a substituting amino acid (Figure 2e). The affected region is highly conserved among different species (Figure 2b). To assess mutant protein stability, I-Mutant 2.0 software was used. Predicted protein stability change upon mutation was estimated as decrease with RI = 3 (reliability index).

The presence of the c.443G>A variant was confirmed by Sanger sequencing (Figure 2c). Data was compared to the published *PPARG* gene sequence NM_015869.4. The variant was submitted to ClinVar and assigned the accession number SCV001622778.

## 5. Discussion

This paper describes the case of a 29-year-old patient harbouring a novel heterozygous *PPARG* mutation c.443G>A, in the DNA-binding domain of the PPARγ protein. The discovery of the missense variant, resulting in glycine to glutamic acid substitution at position 148 is a new one and broadens the spectrum of disease-causing genetic factors contributing to familial partial lipodystrophy type 3.

The PPARγ nuclear receptor is activated by a number of coactivators and corepressors that can either stimulate or inhibit its function [30]. It plays a crucial role in lipid and glucose homeostasis, and as such, any disruptions in its functioning can result in the manifestation of metabolic disturbances such as insulin resistance, diabetes mellitus, hypertriglyceridemia or hepatic steatosis.

It is interesting how alterations in separate genes can result in a similar phenotype, as illustrated by the resemblance between the two most common types of congenital lipodystrophy (FPLD2 and FPLD3). It is known that mutations in *PPARG* disrupt the differentiation of adipocytes, while *LMNA* mutations lead to their premature apoptosis. Numerous transcriptional factors regulate lamin and PPARγ activity. It was found that SREBP1 (sterol regulatory element-binding transcription factor 1) binds to Lamin A, but also activates expression of many genes, including *PPARG*. It is suspected that abnormal amount of prelamin A, as a consequence of *LMNA* mutation, could significantly decrease the pool of active SREBP1, which may also affect *PPARG* expression. However, the precise molecular mechanism of invalid prelamin A/SREBP1 binding remains unclear [11,31]. The overlap between the FPDL2 and FPLD3 phenotypes is certainly associated with its effects on the various stages of adipogenesis. The relationship between *LMNA* and *PPARG* genes remains unclear. In the 29-year-old index patient, the first hallmark of ongoing disease was severe dyslipidemia, with a triglyceride level reaching 1700 mg/dL and total cholesterol of 400 mg/dL. Interestingly, a high triglyceride level (360 mg/dL) was also detected in the proband’s son, aged 3 years, which strongly indicated a genetic background of the disease. Hypertriglyceridemia is commonly reported with varying degrees of severity in FPL patients—some present very high TG levels (>500 mg/dL), while others only demonstrate a slight elevation. Most authors propose that the severity of lipid disturbances translates directly into body fat reduction, which is more pronounced in women. Moreover, Lazarte et al. suggest that the risk of severe hypertriglyceridemia and consequent pancreatitis in FPLD2 depends on the co-occurrence of diabetes [32]. Our index patient and her mother also demonstrate severe hypertriglyceridemia with features of high cellular insulin resistance, which supports the findings of other studies. This impaired lipid metabolism in the patient resulted in the development of hepatic steatosis. As the blood test is one of the basic laboratory analyses performed routinely, often these extremely high TG levels are the first sign of fatty tissue disorder. Furthermore, this should be a relevant indication to control possible IR and implement early treatment by the clinician.

The main physical feature characterizing all familial partial lipodystrophies is gradual subcutaneous adipose tissue loss from the extremities, starting at puberty. The other characteristics of specific FPLD subtypes may be poorly expressed or unnoticeable. Physical examination of our index patient revealed atrophy of fat tissue on the upper and lower limbs, waist and chest region with excess fat deposits on the face and neck and hirsutism. In this cases, a diagnosis of partial lipodystrophy was made, with no further phenotypic categorisation between FPLD2 and FPLD3, which is sometimes troublesome. In general, FPLD3 individuals have less extensive adipose tissue loss, more severe and earlier occurring acanthosis nigricans, hepatic steatosis, PCOS, hirsutism, hypertension, diabetes type 2 and greater biochemical insulin resistance [1]; patients with FPLD3 are more likely to present severe clinical and biochemical disturbances, disproportionate to the lipodystrophy extent compared to FPLD2 [33]. It should be also noted that there is no firm diagnostic criteria for lipodystrophy, because of its rarity and high degree of genetic heterogeneity [34]; as such, it can easily go unrecognized, or be misdiagnosed as developing metabolic syndrome [35]. One of the essential hallmarks of congenital lipodystrophy, distinguishing it from uncontrolled diabetes mellitus or thyrotoxicosis, is the inability to recover a proper fat distribution [34]. Further identification of a specific lipodystrophy subtype requires genetic testing.

Molecular diagnosis of FPLD can be based on a single or panel gene sequencing, or even more comprehensive technology like whole-exome sequencing (WES) or whole-genome sequencing (WGS). Considering that the genetic background of congenital lipodystrophies are known only to a small extent, the use of high-throughput technologies seems to be justified. In the present study, exon sequencing of the *LMNA* and *PPARG* genes was performed simultaneously based on a custom NGS panel. The conducted analysis revealed no pathogenic variants within the *LMNA* gene; however the missense heterozygous variant c.443G>A was noted in exon 4 of *PPARG*. The detected mutation results in a glycine to glutamic acid substitution at position 148 (p.Gly148Glu; G148E). Sanger sequencing confirmed the presence of the same variant in the proband’s son and symptomatic mother.

The substitution results in the formation of an additional side chain (-CH_2_CH_2_-COOH) within the DNA-binding domain of PPARγ, which is highly conserved among other receptors and species. Figure 2b presents the conducted multiple sequence alignment. Combined computational analysis based on 15 predictor tools (BayesDel_addAF, DEOGEN2, EIGEN, FATHMM, LIST-S2, LRT, M-CAP, MVP, MutPred, MutationAssessor, MutationTaster, PolyPhen2, PROVEAN, PrimateAI and SIFT) on Varsome (https://varsome.com) classified the above variant as pathogenic, with no benign predictions from any of them. Mutant protein stability estimated by I-Mutant 2.0 software was defined as decreased. The detected aberration cosegregates with FPLD phenotype in the proband’s family.

Cases of DNA-binding domain mutations and their unequivocal effect on PPARγ receptor effectiveness have been described in the literature. For example, heterozygous PPARG mutations (C114R, C131Y, C162W) located within DBD, inhibit wild-type receptor activity by a dominant negative mechanism [36]. In turn, Visser et al. showed that pathogenic Y151C displayed impaired DNA-binding capacity and hence reduced transcriptional activity compared with wild type PPARγ [37]. Ludtke et al. demonstrated that a receptor with a novel C190S variant has significantly lower ability to activate the reporter gene compared to wild-type protein, without any observations of dominant negative effects [38]. Based on the available data, it is reasonable to assume that G148E plays an important role due to crucial localization within the DNA-binding domain; its presence can potentially lead to invalid receptor attachment and decreased PPARG transcriptional activity.

To the best of our knowledge this is the first time that the c.443G>A, p.(Gly148Glu) variant has been reported. The presented computational analysis, variant cosegregation and literature review support our hypothesis about the pathogenicity of G148E. However, functional testing is needed to confirm the pathogenicity of this variant.

## 6. Materials and Methods

Due to a clinical suspicion of lipodystrophy the patient was referred to the Department of Genetics. Genomic DNA was automatically extracted from peripheral blood lymphocytes using MagCore Genomic DNA Whole Blood Kit (RBC Bioscience, New Taipei City, Taiwan), according to the manufacturer’s instructions. Quantitative and qualitative assessment of extracted DNA was performed using aNanoDrop 2000 spectrophotometer (Thermo Fisher Scientific, Waltham, MA, USA). Sample sequencing was performed on MiniSeq sequencer (Illumina, San Diego, CA, USA), with the use of custom designed lipid NGS panel covering all exons and the exon-intron boundaries of 21 genes (*ABCA1*, *ABCG5*, *ABCG8*, *APOA5*, *APOB*, *APOC2*, *APOE*, *CYP7A1*, *GPIHBP1*, *LCAT*, *LDLR*, *LDLRAP1*, *LIPA*, *LMF1*, *LMNA*, *LPL*, *PCSK9*, *PPARG*, *SCAP*, *SREBF2*, *STAP1)*, including the *LMNA* and *PPARG* gene. Probes for the targeted regions were designed using Illumina Design Studio (2× 150 base pair read length in paired-end mode). Libraries were prepared using TruSeq Custom Amplicon Low Input Library Prep Kit according to the manufacturer’s protocol (Illumina). The PhiX library was combined with a prepared library and used as a sequencing control. Identification, annotation and classification of disease-relevant variants was conducted by Variant Studio 3.0 (Illumina). The presence of identified c.443G>A variant was confirmed by bidirectional Sanger sequencing on 3500 Series Genetic Analyzer (Applied Biosystems, Waltham, MA, USA). DNA Variants Analysis was performed using Mutation Surveyor V5.1.0 software (SoftGenetics, State College, PA, USA). No pathogenic variants were found in other genes.

Informed consent was obtained from all participants or their legal guardian. The study was approved by the ethics committee of the Polish Mother’s Memorial Hospital Research Institute (No. 15/2016 from 12 January 2016).

## 7. Conclusions

Our presented literature review illustrates the genetic heterogeneity of congenital lipodystrophies and their wide spectrum of severe metabolic complications. Due to the overlapping clinical symptoms they can be easily misdiagnosed as early onset insulin resistant diabetes mellitus, persistent hypertriglyceridemia, hepatic steatosis, PCOS or hepatosplenomegaly. Description of a new cases is highly needed both in terms of understanding the disease pathology from clinical point, but also increasing the awareness of rare diseases, such as congenital lipodystrophy syndromes. Apart from the relevant role of an experienced clinician, a significant contribution of genetic diagnosis cannot be omitted. Searching for new genetic backgrounds brings us closer to better understanding the origin of metabolic consequences, therefore improving the diagnostic and treatment pathways. Further analyses of known and candidate genes implicated in familial partial lipodystrophy are highly needed.

## Figures and Tables

**Figure 1 diagnostics-12-01122-f001:**
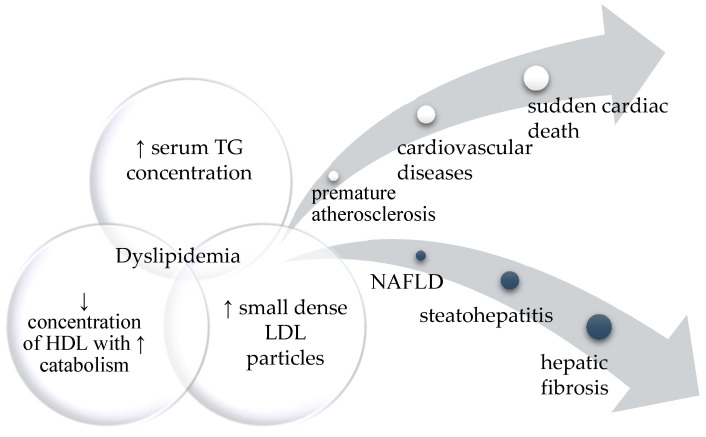
The diagram represents the main components of dyslipidemia and its possible health consequences. If left untreated, it can affect different organs leading to severe cardiovascular disease or various degrees of fatty liver disease.

**Figure 2 diagnostics-12-01122-f002:**
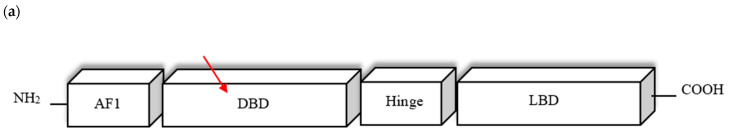
(**a**) Schematic presentation of PPARγ domain organization, showing the location of novel G148E mutation. (**b**) Multiple sequence alignment of the amino acid at position 148 of the PPARγ protein from various species (CHICK—*Gallus gallus*; VOMUR—*Vombatus ursinus*, MOUSE—*Mus musculus*, BOVIN—*Bos Taurus*, PIG—*Sus scrofa*, CANLF—*Canis lupus familiaris*, MACMU—*Macaca mulatta*, HUMAN—*Homo sapiens*) using Jalview 2.11.0 and Clustal Omega 1.2.4. The conserved glycine amino acid at position 148 is indicated by red frame. (**c**) Sequence chromatogram showing c.443G>A variant of both forward and reverse strand. (**d**) Schematic amino acid structure of zinc finger I. Substitution G→E at position 148 is marked in red. (**e**) The homology model of wild type (i) and mutant (ii) PPARγ protein. The red arrow indicates the position of amino acid substitution. Homology modelling was conducted using SWISS-MODEL. PPARγ protein template model was downloaded from the uniprot.org. Both protein models were compared in PyMOL 2.5.2 software. (**f**) Result of mutant protein stability assessment conducted by I-Mutant 2.0. Predicted protein stability change upon mutation was estimated as decrease with RI = 3 (reliability index), where 10 being the highest. The tool uses data derived from ProTherm [29].

**Table 1 diagnostics-12-01122-t001:** Subdivisions of Familial Partial Lipodystrophy.

Type	Major Genetic Background	Manner of Inheritance	OMIM Number	Observed Phenotype
FPLD type 1, Kobberling	unknown/polygenetic origin	-	%608600	Loss of subcutaneous fat from the limbs with truncal obesity Reduction of gluteal AT Normal or increased facial and neck AT
FPLD type 2, Dunnigan	*LMNA*	dominant	#151660	Loss of subcutaneous fat from the limbs and trunk Reduction of gluteal AT Excess fat accumulation in the face and neck Increased muscularity
FPLD type 3	*PPARG*	dominant	#604367	Loss of subcutaneous fat from the lower limbs Normal or increased abdominal, facial and neck AT Increased muscularity
FPLD type 4	*PLIN1*	dominant	#613877	Loss of subcutaneous fat primarily in gluteal and lower limb regions Muscular appearance
FPLD type 5	*CIDEC*	recessive	#615238	Lack of AT on limbs and gluteal region Presence of visceral, neck and axillary fat pads Increased muscularity
FPLD type 6	*LIPE*	recessive	#615980	Reduced lower limbs subcutaneous fat In some patients abnormal fat accumulation in the back and axillae
FPLD type 7	*CAV1*	dominant	#606721	Absence of AT over entire body except buttocks, hips and thighs

AT—adipose tissue.

**Table 2 diagnostics-12-01122-t002:** The characterisation of *PPARG* domains and their functions.

Structural Domains	Functional Domains	Role	Degree of Conservation
N-terminus A/B	AF-1 Ligand-independent transactivation function 1	Regulates the ligand-independent transcriptional *PPARG* activity	Poorly conserved
C	DBD DNA-binding domain	Binding PPARγ to the promoter region of the targeted genes	Highly conserved
D	HINGE flexible hinge region	Involved in interaction with coactivators and corepressors	Poorly conserved
C-terminus E/F	LBD Ligand binding domain AF-2 Ligand-dependent transactivation function 2	Regulates the ligand-dependent transcriptional *PPARG* activity; Responsible for dimerization with RXR	Highly conserved

**Table 3 diagnostics-12-01122-t003:** Serum lipid concentration values of described family members.

	TC [mg/dL]	LDL [mg/dL]	HDL [mg/dL]	TG [mg/dL]
Index case	299	111	25	870
Mother	232 (* 573)	102	54	382 (* 1989)
Son	141	88	28	360

* highest reported value.

## Data Availability

The data of novel variant that we describe in the manuscript was submitted to ClinVar database with the accession number SCV001622778.

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
