# Peer review of "Familial Partial Lipodystrophy—Literature Review and Report of a Novel Variant in PPARG Expanding the Spectrum of Disease-Causing Alterations in FPLD3"

_diagnostics, 2022, doi:10.3390/diagnostics12051122_

Round 1

Reviewer 1 Report

The authors present a new PPARG variant carried by a woman, her mother and son with a suspicion of partial familial lipodystrophy (PFLD). They performed NSG sequencing of 23 gene including LMNA and PPARG, which are the main genes involved in PFDL2 and PFLD3, on these 3 related. The clinical description is complete and sequencing data analysis also. The authors also include a literature review part to better introduce the case reported.

Minor comments:

  • Line 98 and 100 mutations should be mentioned according to HGVS nomenclature recommendations
  • The authors should check all the references because there are some mistakes, for example: line 125 the reference [17] is not about PPARG mutations; line 150 and 168 the ref [20] is not the right one. It is necessary to check the entire bibliography
  • When the authors cite a literature review they should cite the specific article that illustrate the point (for example ref [9] line 130)
  • Line 266 the accession number assigned by Clinvar is wrong
  • A space is missing line 324 between consequent and pancreatitis and line 363 between within and the
  • In the material and method, the author should mention the 23 genes of the panel

Author Response

Dear Reviewer,

Thank you for giving us the opportunity to submit a revised draft of our manuscript to special edition “Recent Advances in the Diagnosis of Metabolic Disorders” of Diagnostics. We highly appreciate the time and effort to providing your valuable feedback on our manuscript. We are pleased that the paper has been positively evaluated.

We have been able to incorporate changes to reflect most of the suggestions. We have heighted all revisions within the manuscript using the “Track Changes”. Bibliography mistakes, that you have mentioned, have been corrected and the entire bibliography has been rechecked. The given accession number is correct and will be visible in the ClinVar database upon publication of the article. ClinVar, for the protection of a new researchers’ finding, provides the option to exclude a submission from public access until the data is published.

Reviewer 2 Report

Some comments for Authors:

In general, nowadays we are trying to use the term "lipodystrophy syndromes" in plural, as opposed "lipodystrophies".

Not sure about the Title

The structure of the article is a little bit unusual - the description of a clinical case is called "Results" and "Methods" are in the very end, but if it's in accordance with a typical structure of the articles in the journal, then fine.

You mention (line 395) that NGS panel covering all exons and the exon-intron boundaries of 23 genes - however there is no information about other 21 genes except LMNA and PPARG, it should be added with the indication that no pathogenic variants were found in them.

I am not sure about the Conclusion. I would think that identifying a new PPARG mutation in a patient with FPL3 brings more to clinicians in terms of understanding the pathology from the clinical point of view, increases the awareness about the disease and helps to recognize it among patients with insulin resistance and diabetes. Your conclusion, however (lines 411-414) is not anyhow connected with the clinical case you present, maybe with review a little bit, but I would still recommend to rephrase it.

In my opinion in the Review part more attention should be payed to FPL3, especcially clinical part, and maybe less to general information about different FPLs and FPL2, as your patient is a condidate for FPL3. It would be good to see the comparison of this clinical case to previously published one s with FPL3 (there are not too many).

In discussion the author's focused on clinical similarities between FPL2 and FPL3, however for a specialist with some experience with lipodystrophy syndromes the differences are quite clear and can be noticed and there were works comparing FPL2 and FPL3.

(12) Its estimated prevalence is as low as 1 in 1 million. - check the prevalence, it's different for different forms, and for FPL it's more, about 2.84 per million - Chiquette et al., 2017

(12) The deficiency of 
(13)metabolically active adipose tissue is closely linked with a wide range of metabolic complications, 
(14) such as insulin resistance, type 2 diabetes mellitus, dyslipidemia with severe hypertriglyceridemia,
(15) hypertension or hepatic steatosis. - It is incorrect to reffer to lipoatrophic diabetes as "type 2 diabetes mellitus", it is a different type of diabetes

(34) Inherited lipodystrophies represent a heterogenous group of metabolic disorders, whose back- 
(35) ground remains unclear. - That is not exactly relevant to the case. In about 50% of cases of inherited lipodystrophy we can't find the mutations in the known candidate genes, if that's what the author means.

(45) Lipodystro-
(46) phies can be partial or generalized in distribution [2]. - that phrase is not very clear: based on the deggree of fat loss we distinguish partial and generalized lipodystrophy

(53) The estimated prevalence of CGL is approximately 1 in 10 million and for 
(54) FPLD 1 in 1 million [3]. - careful about these numbers, there are other expectations about FPL (see above) and also expectations of low diagnostics

(62) Familial Partial Lipodystrophy is a rare genetic disorder characterized by selective 
(63) loss of adipose tissue in the extremities and gluteal region, without any change in ab- 
(64) dominal and visceral fat. - Abdominal fat can be absent in FPL2, or increased in FPL1 and some other types, so it is better to correct this sentence

(66) with a wide range of metabolic complications, such as insulin resistance, type 2 diabetes 
(67) mellitus, - again, lipoatrophic diabetes, not T2DM

(69) The extent of fat loss often determines the severity of metabolic 
(70) consequences. -That is questionably, better provide some proof of what you meant here. In FPL it can be the opposite - the more ectopic fat, the worse metabolic complications. If you ment that GL is generally associated with worse metabolic complications than PL- that's differnet.

(71) There are seven subtypes of FPLD (characterised in Table 1) caused by mutations in - However FPLD type 1 (the most common and questionable type) genetic origin remains unknown, so you need to rephrase that
(72) various genes (LMNA, PPARG, PLIN1, CIDEC, LIPE, AKT2 or CAV1); the condition can be
(73) inherited in either a dominant or recessive manner. - but mostly dominant, it's better to highlight it

A bit more clinical details of the index case and the mother would be nice - were skin folds measurements done? any imafing? DXA? body impedancemetry? 

Author Response

Dear Reviewer,

Thank you for giving us the opportunity to submit a revised manuscript to special edition “Recent Advances in the Diagnosis of Metabolic Disorders” of Diagnostics. We highly appreciate the time and effort to providing your valuable feedback on our manuscript. We have been able to incorporate changes to reflect most of the suggestions. We have heighted all revisions within the manuscript using the “Track Changes”. We would like also further respond to certain comments:

Comment:

In my opinion in the Review part more attention should be payed to FPL3, especially clinical part, and maybe less to general information about different FPLs and FPL2, as your patient is a condidate for FPL3. It would be good to see the comparison of this clinical case to previously published one s with FPL3 (there are not too many).

In discussion the author's focused on clinical similarities between FPL2 and FPL3, however for a specialist with some experience with lipodystrophy syndromes the differences are quite clear and can be noticed and there were works comparing FPL2 and FPL3.

Response:

Thank you for your opinion. The presented literature review is focused mainly on FPLD type 2 and 3, as this was the scope of clinical consideration in our case (line 58). We have presented not only a case report but also a wider literature review to introduce the reader to the topic and to gather previous knowledge in this field. Many authors have pointed out numerous phenotypical similarities, in both types, occurring with varying severity, so it is difficult to define a hard line between this two. As FPLD2 is better described, we referred the found information about FPLD3 to that reported in type 2 (line 152-157, 171-173, 193-199). Since the genetic aspect and molecular diagnosis of rare diseases is our main area of interest, we have focused on those fields referring to the other publications, that was described from the line 376.

Comment:

Its estimated prevalence is as low as 1 in 1 million. - check the prevalence, it's different for different forms, and for FPL it's more, about 2.84 per million - Chiquette et al., 2017

Response:

Thank you for your suggestion. Citing a source Chiquette E et al. Estimating the prevalence of generalized and partial lipodystrophy: findings and challenges. Diabetes Metab Syndr Obes. 2017;10:375-383:

“The prevalence range of all LD across all EMR databases was 1.3–4.7 cases/million. For the adjudicated Quintiles search, the estimated prevalence of diagnosed LD was 3.07 cases/million (95% confidence interval [CI], 2.30–4.02), 0.23 cases/million (95% CI, 0.06–0.59) and 2.84 cases/million (95% CI, 2.10–3.75) for generalized lipodystrophy (GL) and partial lipodystrophy (PL), respectively. For all literature searches, the prevalence of all LD in Europe was 2.63 cases/million (0.96 and 1.67 cases/million for GL and PL, respectively).”

From the data quoted above, it can be seen that the author estimated the prevalence for Partial Lipodystrophy, which is Familial Partial Lipodystrophy and  Acquired Partial Lipodystrophy. In the line 12 we are only referring to FPLD, so we are not sure if we can use this value. The described prevalence 1/1.000.000 was extracted from the Orphanet database but they also point out that given value may be underestimate. There are rather general lipodystrophy syndromes prevalence described in the literature. This is probably due to the fact that the disease is still underdiagnosed and the actual number of patients is constantly being discovered.

Comment:

A bit more clinical details of the index case and the mother would be nice - were skin folds measurements done? any imafing? DXA? body impedancemetry?

Response:

We have included all available information about the patient in the manuscript. To the best of our knowledge patient did not have any imaging studies.